# Using tooth enamel microstructure to identify mammalian fossils at an Eocene Arctic forest

**Jaelyn J. Eberle**[1]◉*, **Wighart von Koenigswald**[2]◉, **David A. Eberth**[3]◉

**1** University of Colorado Museum of Natural History and Department of Geological Sciences, University of Colorado, Boulder, Colorado, United States of America, **2** Institute of Geosciences (Section Palaeontology) at the Rheinische Friedrich-Wilhelms University, Bonn, Germany, **3** Royal Tyrrell Museum, Drumheller, Alberta, Canada

◉ These authors contributed equally to this work.
* Jaelyn.Eberle@Colorado.edu

**Data Availability Statement:** All relevant data are within the manuscript.

**Funding:** The 2010 fieldwork that resulted in the discovery of the tooth fragments at the Strathcona Fiord Fossil Forest was supported by National

## Abstract

Lower Eocene (Wasatchian-aged) sediments of the Margaret Formation on Ellesmere Island in Canada's High Arctic preserve evidence of a rainforest inhabited by alligators, turtles, and a diverse mammalian fauna. The mammalian fossils are fragmentary and often poorly preserved. Here, we offer an alternative method for their identification. Among the best preserved and extensive of the Eocene Arctic forests is the Strathcona Fiord Fossil Forest, which contains permineralized *in situ* tree stumps protruding from a prominent coal seam, but a paucity of vertebrate fossils. In 2010 and 2018, we recovered mammalian tooth fragments at the fossil forest, but they are so incomplete as to be undiagnostic by using their external morphology. We used a combination of light microscopy and SEM analysis to study the enamel microstructure of two tooth fragments from the fossil forest—NUFV2092B and 2092E. The results of our analysis indicate that NUFV2092B and 2092E have *Coryphodon*-enamel, which is characterized by vertical bodies that manifest as bands of nested chevrons or treelike structures visible in the tangential section under light microscopy. This enamel type is not found in other mammals known from the Arctic. Additionally, when studied under SEM, the enamel of NUFV2092B and 2092E has rounded prisms that open to one side and are surrounded by interprismatic matrix that is nearly parallel to the prisms, which also occurs in *Coryphodon* enamel, based on prior studies. The tooth fragments reported here, along with some poorly preserved bone fragments, thus far are the only documented vertebrate fossils from the Strathcona Fiord Fossil Forest. However, fossils of *Coryphodon* occur elsewhere in the Margaret Formation, so its presence at the fossil forest is not surprising. What is novel in our study is the way in which we identified the fossils using their enamel microstructure.

## Introduction

Lower Eocene (Wasatchian-aged) strata of the Margaret Formation, Eureka Sound Group on Ellesmere Island, Nunavut preserve evidence of lush mixed conifer-broadleaf rainforests

Science Foundation grant ARC-0804627 to JJE (https://www.nsf.gov/). The 2018 fieldwork by JJE that recovered additional tooth fragments was made possible by support from the National Science Foundation Grant No. DRL-1713552 awarded to WGBH Educational Foundation (July 18, 2017) for the project entitled "Polar Extremes: Enhancing Experiential Digital Learning." Enamel microstructure analysis of the fossils by JJE and WvK at the University of Bonn, Germany was supported by an award to JJE from the Deutscher Akademischer Austauschdienst German Academic Exchange Service (DAAD). The funders had no role in study design, data collection and analysis, decision to publish, or preparation of the manuscript.

**Competing interests:** The authors have declared that no competing interests exist.

inhabited by alligators, turtles, birds, and at least 25 mammalian genera [1]. First discovered in 1975 near the head of Strathcona Fiord [2], the vertebrate fossils in the Eureka Sound Group are few, fragmentary and weather-worn, which can make it challenging to identify them. By far the most abundant and diverse vertebrate fossils have come from the Margaret Formation cropping out on the Matthew peninsula between Bay and Strathcona fiords [3] (Fig 1).

Nearly a century before the first discovery of Eocene vertebrate fossils on Ellesmere, Eocene Arctic forests were first documented when Sergeant D.L. Brainard, a survivor of the ill-fated Greely expedition of 1881–1883, discovered petrified logs on northeastern Ellesmere Island [4]. Among the best preserved, extensive, and photogenic of the Eocene Arctic fossil forests is the Strathcona Fiord Fossil Forest, which preserves permineralized *in situ* tree stumps protruding from a prominent coal seam (Figs 1 and 2). The tree stumps are large, with diameters ranging from 40 cm to over a meter, and closely spaced, indicating a dense forest comparable to today's cypress swamps in the southern United States [5]. Eocene Arctic paleotemperature estimates using multiple proxies suggest a mean annual temperature (MAT) of 5–17°C, with winters above freezing and summer temperatures above 20°C [6–8]. Further, the Eocene Arctic rainforests had high mean annual precipitation and humidity, comparable with today's temperate rainforests along the North American west coast [8].

In the summers of 2010 and 2018, JE, DE and team recovered tooth fragments belonging to a mammal and small, undiagnostic bone fragments at the Strathcona Fiord Fossil Forest. As far as we are aware, these are the first vertebrate fossils documented from this fossil forest. Most fossil mammals can be identified to genus and often species by dental morphology. However, the tooth fragments from the Strathcona Fiord Fossil Forest are so incomplete as to be undiagnostic by using their external morphology. Therefore, we analyzed the tooth enamel microstructure to assist us in identifying the fossils. Mammalian prismatic enamel, the hardest

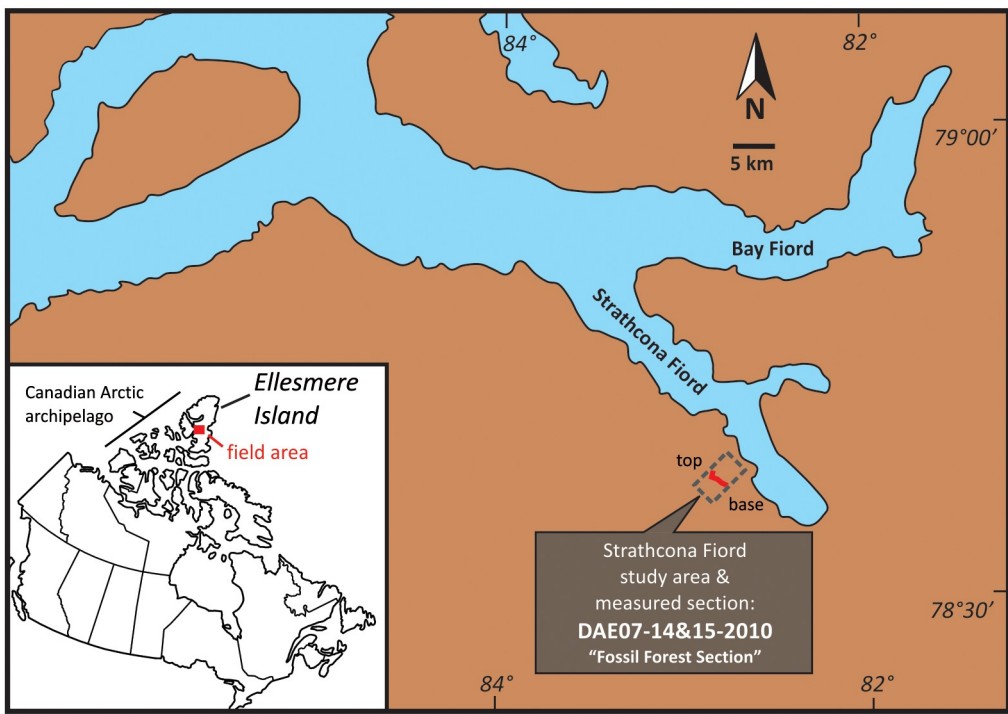

**Fig 1. Map of Canada (inset) and satellite image showing location of the Strathcona Fiord study area and measured section.**

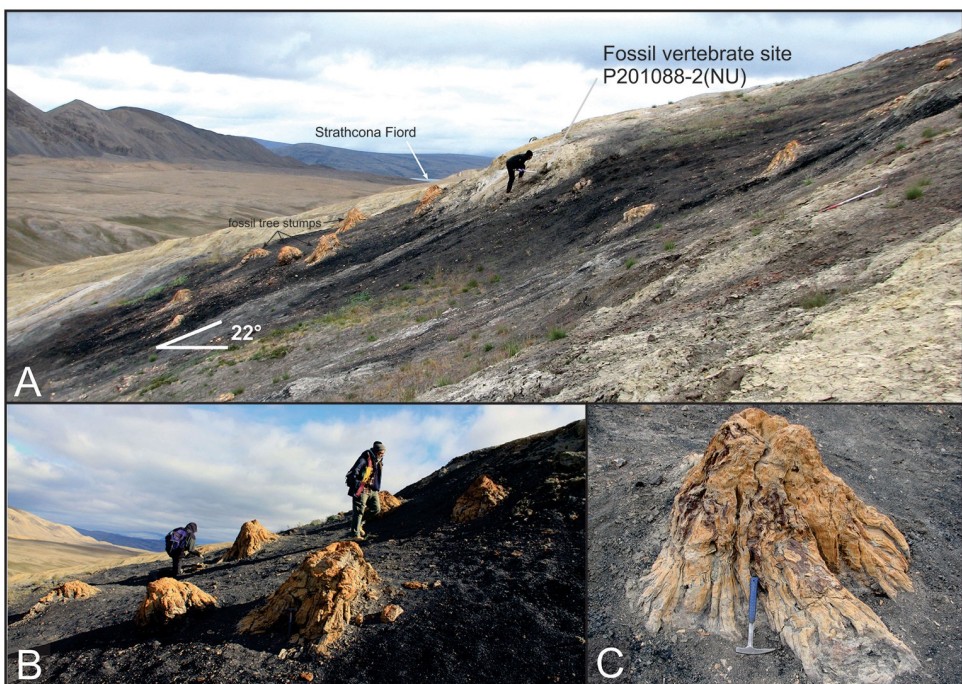

**Fig 2. Images of the Strathcona Fiord Fossil Forest and fossil vertebrate site P201088-2(NU).** (A) P201088-2(NU) is less than a meter stratigraphically above the coal containing the fossil forest; (B) and (C) show close-ups of petrified tree stumps in the fossil forest.

and most resistant material in the body, consistently differs in its microstructure among clades of Perissodactyla (odd-toed ungulates; [9]), Rodentia [10], Proboscidea [11], and other groups. Here, we demonstrate that the tooth fragments from the Strathcona Fiord Fossil Forest can be identified to genus based on their enamel microstructure. In so doing, we (1) document the first fossil mammal from the Strathcona Fiord Fossil Forest; (2) help refine the age and correlation of this site within the context of the Margaret Formation elsewhere on Ellesmere Island; and (3) underscore the utility of tooth enamel microstructure in identifying mammalian tooth fragments that cannot be identified by traditional paleontologic means.

## Geologic setting

The tooth fragments described below were recovered from Locality P201088-2(NU) on central Ellesmere Island, just south of 78˚ 40' N latitude along the southwestern coastal region of Strathcona Fiord (Fig 1). The general area, referred to as "Strathcona Fiord" [1], includes the first two fossil vertebrate sites that were discovered in the Canadian Arctic (Ellesmere Island) by Dawson and her colleagues in the 1970s [2], approximately 10–12 km southeast and northeast of locality P201088-2(NU).

The sharp-based sandstone that hosts locality P201088-2(NU) occurs less than a meter above the top of an approximately 2 m-thick coal that preserves an assemblage of *in situ* permineralized tree stumps and root balls coined the Strathcona Fiord Fossil Forest ([5, 12]; Figs 2 and 3).

Vertebrate fossils in the Strathcona Fiord region produce from the Eureka Sound Group, which consists of four formations in this area. In ascending order, these are the Mount Bell, Mount Lawson, Mount Moore, and Margaret formations [13, 14]. The stratigraphic section

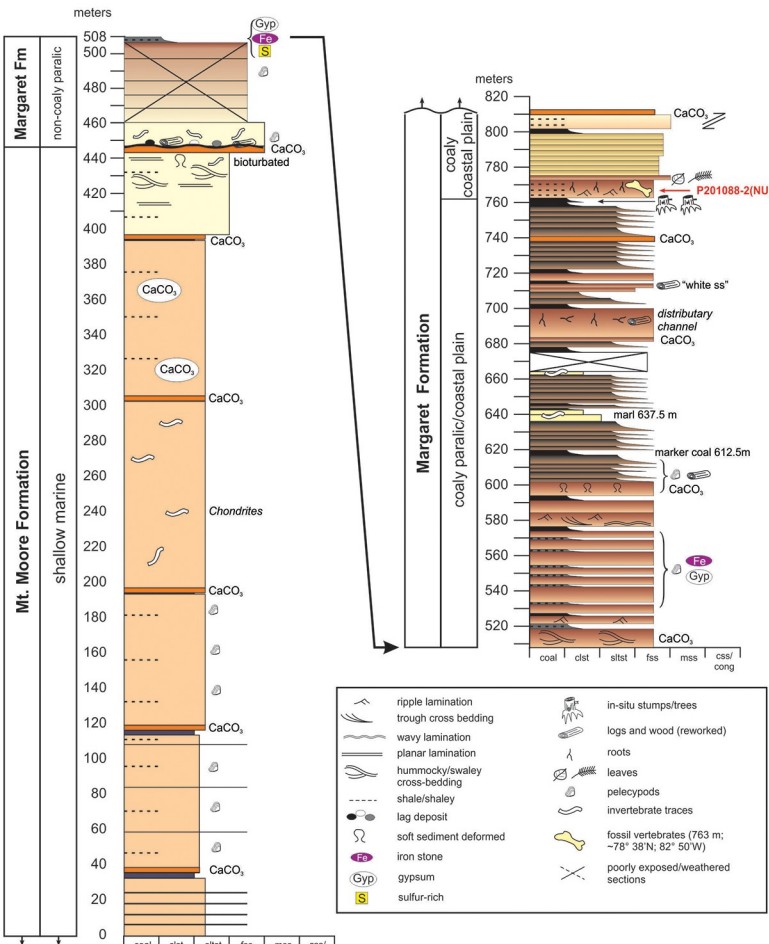

**Strathcona Fiord Section:** DAE July14–15, 2010

**Fig 3. Strathcona Fiord Section showing stratigraphic level of fossil vertebrate locality P2011088-2 (NU) and Strathcona Fiord Fossil Forest in the coal directly below it.** Sedimentology is described in the text.

that we examined and measured at Strathcona Fiord (Fig 3) consists of the uppermost exposures of the fine-grained marine Mount Moore Formation, overlain by non-coaly to coaly, paralic-to-non-marine deposits of the lower Margaret Formation [1, 13–15]. We place locality P201088-2(NU) approximately 316 m above the base of the multi-kilometer-thick Margaret Formation. Based on the thickness data of [13] and [14], we regard this position as occurring in the lower portion of the Margaret Formation. Beds in this portion of the section are exposed along an extensive north-south trending ridge, and dip steeply (25˚– 35˚) to the west. Lastly, we note that during stratigraphic measurement and examination in our study area, we identified errors in formation identification on the geologic map [15]. Specifically, these authors erroneously mapped the Strathcona Fiord Fossil Forest and its thick coal as part of the Mount Lawson Formation, which instead is a Paleocene-aged marine mudstone succession in the lower half of the Eureka Sound Group [13].

Strathcona Fiord fossil vertebrate sites have previously been interpreted as early Eocene (Wasatchian) in age [1–3, 14]. Based on these interpretations and stratigraphic patterns [14], we suggest the age of the Strathcona Fiord Fossil Forest and locality P201088-2(NU) is also

Wasatchian, equivalent in time to the lower Margaret Formation interval at Bay Fiord (~25 kilometers to the north-northeast).

Locality P201088-2(NU) occurs just above the stratigraphic horizon where the Margaret Formation paralic succession (complexly and thinly interbedded marine to non-marine strata) transitions up-section into a 50 m thick succession of strictly non-marine, coastal-plain strata dominated by fine sandstones and minor coals. Exposures of the lower Margaret Formation at Bay Fiord [14] record a similar pattern of overall regression and are characterized by an up-section transition from paralic to coaly coastal plain deposits. The host sandstone for locality P201088-2(NU) exhibits ripple laminae and abundant coalified root traces, the latter indicating that a stable and likely subaerial substrate was present for plant colonization subsequent to deposition of the sandstone. Sparse occurrences of fossil vertebrate fragments are present among mudstone and ironstone intraclasts in the sandstone, suggesting that lower portions of the sandstone may have been deposited as a lag deposit during waning flow. Angiosperm and gymnosperm leaf fragments, and more root traces are common in the uppermost portions of the host sandstone, again suggesting substrate stability between subsequent later-stage sediment accumulation events.

No other intervals were observed lower in section that preserve well-developed associations of 'fossil forests,' ironstone and fossil vertebrate clasts, rooted horizons, fossil leaves, and an absence of marine indicators. Furthermore, the presence of the fossil forest, multiple rooted horizons, leaf fossils, and ironstone and fossil bioclasts (the tooth fragments described below) all suggest forested conditions, exposed substrates, and incipient soil formation in a well-saturated setting subjected to episodic flooding. Accordingly, we interpret this transitional stratigraphic interval as recording an up-section shift from frequently flooded paralic shoreline settings to a relatively more up-dip coastal-plain setting where non-marine conditions prevailed. This paleoenvironmental interpretation matches those previous hypothesized for the Margaret Formation that describe upward-coarsening cycles of interbedded cross-bedded sandstone, siltstone, mudstone, and coal. These are interpreted as proximal delta-front to delta-plain paleoenvironments characterized by abundant shoreline sands, alluvial to estuarine channels, coal swamps, lagoons and bays, and well-forested, low-gradient interfluves [13, 14, 16, 17].

## Materials and methods

Nunavut Fossil Vertebrate (i.e., NUFV) 2092B and 2092E (Fig 4), the tooth fragments analyzed in this study, were collected along with approximately 20 other tooth fragments and several small, weathered bone fragments by JE, DE, and team in July 2010 and by JE in July 2018 at locality P201088-2(NU). The fossils were collected on Class 2 Nunavut Territory Palaeontologist Permits 2010-003P and 2018-02P issued by the Nunavut Department of Culture and Heritage. Detailed coordinates for locality P201088-2(NU) are on file at the Nunavut Department of Culture and Heritage in Iqaluit and the Canadian Museum of Nature (CMN) in Ottawa, Canada.

Given their similarity in thickness and external appearance, and the fact that they were recovered near one another, the tooth fragments probably represent the same taxon and individual. Based upon their thickness, NUFV 2092B and 2092E are from a relatively large mammal. However, their external morphology does not allow us to reliably identify the mammal to which they belong. Therefore, we decided to study the enamel microstructure. To investigate the microstructure of NUFV 2092B and 2092E, three traditional planes of section were studied —the horizontal (or transverse), vertical, and tangential sections (Fig 5), following [18].

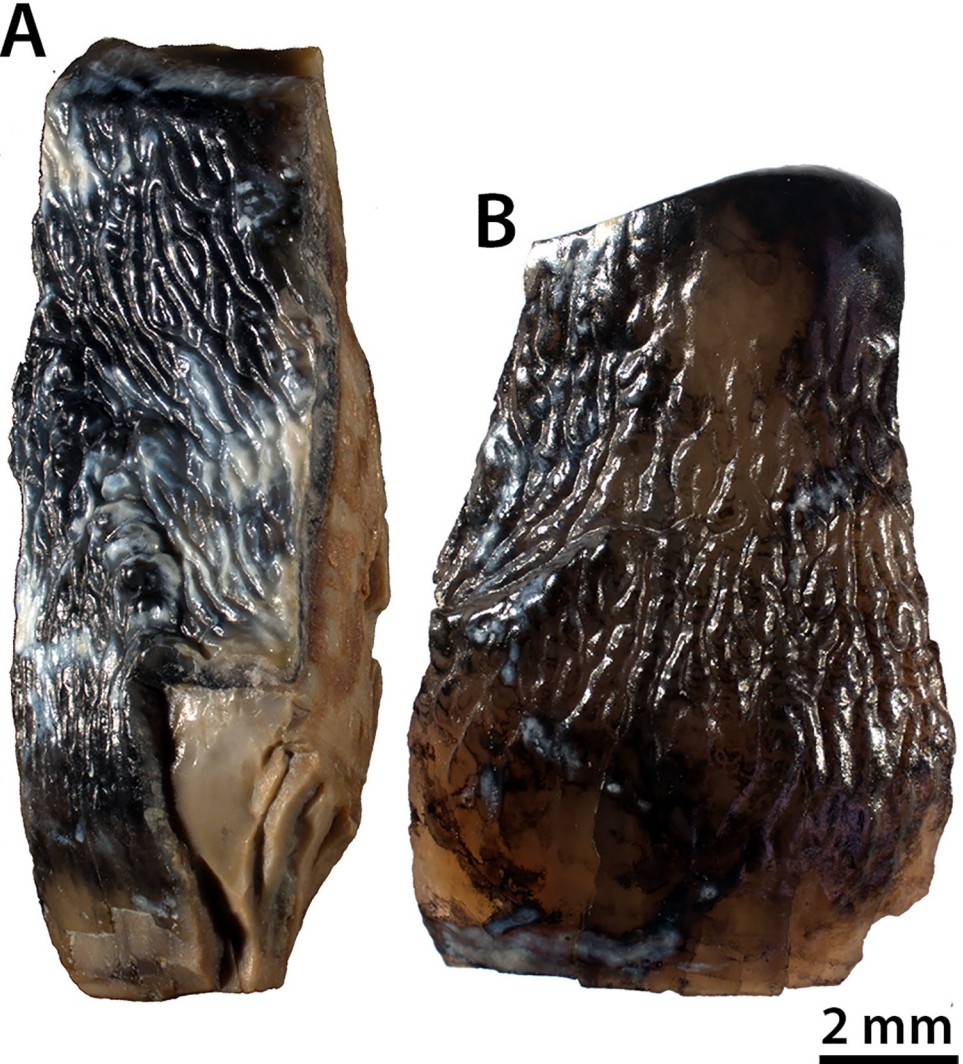

**Fig 4. Tooth fragments from locality P201088-2(NU) at the Strathcona Fiord Fossil Forest site on central Ellesmere Island, Nunavut.** (A) NUFV 2092B. (B) NUFV 2092E.

The techniques for preparing and studying tooth enamel microstructure of large fossil mammals were described by [19] and are summarized here. First, NUFV 2092B and 2092E were studied under a light microscope to orient the specimens and determine the direction of the occlusal surface. The specimens were subsequently embedded in epoxy resin, and oriented so that the desired sections (horizontal, vertical, or tangential) were placed parallel to the resin surface. The embedded specimens were left for 48 hours at room temperature under a fume hood to allow the epoxy to harden. NUFV 2092B was cut using an Isomet© low-speed saw with 0.3 mm blade thickness, to produce horizontal and vertical sections, whereas NUFV 2092E was used for the tangential section. The specimens were ground in three steps. First, they were ground down to the enamel surface using a grinding wheel (grit 240). Next, hand-grinding was done on fine wet sandpaper (grit 800) placed over a glass sheet, and finally the specimens were ground with fine powder grit (grit 1000) mixed with water on a glass plate. Between each of these steps, the specimens were analyzed under a light microscope to ensure

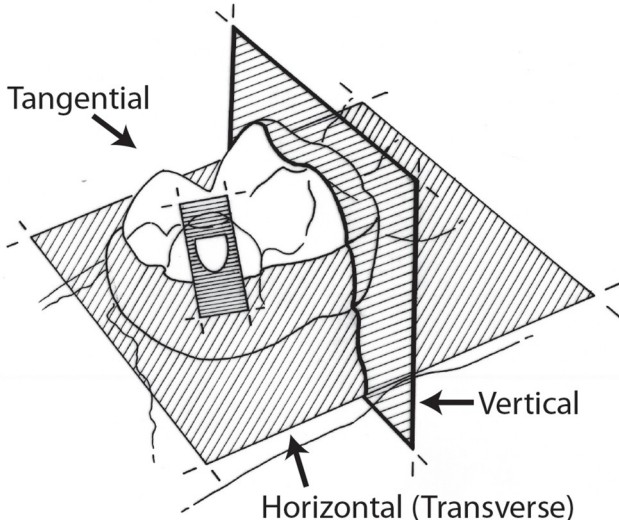

**Fig 5. Orientation of the three traditional planes of section used to study mammalian tooth enamel microstructure from [18].**

that grinding was not too extensive. The ground surfaces were rinsed with water, cleaned in an ultrasonic cleaner, blown dry, and etched with 10% hydrochloric acid (2 normal HCl) for approximately three seconds.

Prior to scanning electron microscope (SEM) analysis, the specimens were sputter-coated with gold or palladium. SEM analysis was conducted on a Camscan MV 2300 instrument in the Palaeontology Section of the Institute of Geosciences and a Cambridge Stereoscan 200 SEM in the Institut für Biodiversität der Pflanzen at the Rheinische Friedrich-Wilhelms University in Bonn, Germany.

In describing the enamel microstructure of NUFV 2092B and 2092E, we follow the hierarchical system of classification of others [20, 21]. Specifically, we define the units by their size and level of complexity. First, we describe the crystallites, the smallest units of enamel that are comprised of fine needles of hydroxyapatite with a diameter of less than 0.5 μm and length of more than 100 μm [22]. Crystallites are visible under high magnification (1500x and higher). Next, we describe the prisms that are made up of bundles of crystallites surrounded by a prism sheath. The size and shape of the prisms differ among clades of mammals. Prisms form at the enamel-dentine junction (EDJ) and grow almost to the outer enamel surface (OES) [23]. They are typically arranged in groups or bands with the same prism orientation. Their orientation defines the various enamel types [9, 24, 25]. The prismatic enamels of mammals often have two or more different enamel types. The most primitive enamel type among placental mammals is radial enamel, in which the prisms' long axes parallel one another and extend radially from near the EDJ towards the OES [18]. More complex enamel types occur in which bands of prisms change their orientations from the EDJ to the OES [20]. Among the most often described enamel types are Hunter-Schreger bands (HSB), which are light and dark stripes often seen under a light microscope. HSB are an optical phenomenon caused by the different prism orientation in alternating bands, forming decussations [9, 24]. HSB occur in the enamel of most large mammals and often are arranged horizontally. However, a few taxa, including rhinocerotoids, have vertical HSB [9, 23, 26]. HSB function as a crack-stopping device [27, 28]. The level above the enamel type is the schmelzmuster, the three-dimensional distribution of

enamel types within the enamel that has both biomechanical and phylogenetic controls. The number of possible combinations of enamel types or schmelzmusters is very large [25].

By studying the horizontal, vertical, and tangential sections of NUFV 2092B and 2092E, we discovered a complex enamel microstructure that was challenging to interpret solely through SEM analysis. Consequently, we also studied NUFV 2092B and 2092E at lower magnification under a light microscope using the light-guide effect described by [29] and more recently by [19]. If light hits an enamel prism approximately perpendicular to its axis, the light is reflected and the prism appears light in color. If, however, the light hits a prism parallel to its long axis, it disappears into the prism, and the prism appears dark. When the source of illumination is changed from one direction to another, bands that were dark in one will be light in the other, and vice versa. The light and dark bands, each comprised of many prisms with the same orientation, are the HSB. We used a combination of the light-guide effect and SEM analysis to study the enamel microstructure of NUFV 2092B and 2092E.

## Results

### Description of NUFV 2092B and 2092E

Based upon thickness, NUVF 2092B and 2092E belong to a relatively large mammal. The shiny outer surface of the enamel is covered by ridges and crenulations that extend vertically and diagonally at a steep angle (Fig 4). Of the diverse mammalian fauna known from the early Eocene Arctic, the pantodont *Coryphodon* is among the largest and best represented by fossils [30], and the enamel on its teeth has vertical ridges and wrinkles [19]. However, these characters are not unique to *Coryphodon*. Many mammals, including brontotheres that also are known from the Eocene Arctic [14, 31, 32], show varying amounts of rugosity and crenulations on the external surface of the enamel.

Others [19] have noted the appearance of vertical stripes in the tangential section of the enamel of *Coryphodon* as well as ridges on the shearing crests of its molars, indicating the presence of vertical structures within the enamel. On the occlusal surface of NUFV 2092B, there are weak ridges, and when the tangential section of the enamel is illuminated from one side, faint vertical light and dark stripes are visible. However, these characters also are not restricted to *Coryphodon*, but occur in rhinocerotoids [23] and some extinct South American mammals including pyrotheres and astrapotheres [29, 33].

*Coryphodon*, however, is characterized by a unique and complex enamel microstructure coined *Coryphodon*-enamel [19]. NUFV 2092B and 2092E were analyzed at low and high magnifications under both a light microscope and scanning electron microscope (SEM) to determine whether *Coryphodon*-enamel is present. Below, we describe the crystallites, prisms, and schmelzmuster of the NUFV specimens, and compare them to the enamel microstructure of *Coryphodon*, rhinocerotoids and other large mammals known from the Eocene Arctic.

### Crystallites and prisms

The cross sections of prisms are best seen in the tangential section (Fig 6A and 6B). Individual crystallites are visible at high magnification, where they appear as fine needles making up the prisms and interprismatic matrix (IPM) (Fig 6B). The crystallites in the prisms appear approximately parallel to those comprising the IPM, which is found in the enamel of *Coryphodon* [19]. Most of the prisms in NUFV 2092E are rounded and have an open prism sheath, although there is some variability in shape, with some prisms being more oblong and narrowing towards the top. The prisms range in diameter from approximately 4–7 μm and the prism sheaths are open towards the base of the tooth or to one side. A horseshoe-shaped prism sheath was noted for *Coryphodon* [19], although it is not restricted to the genus. Although close

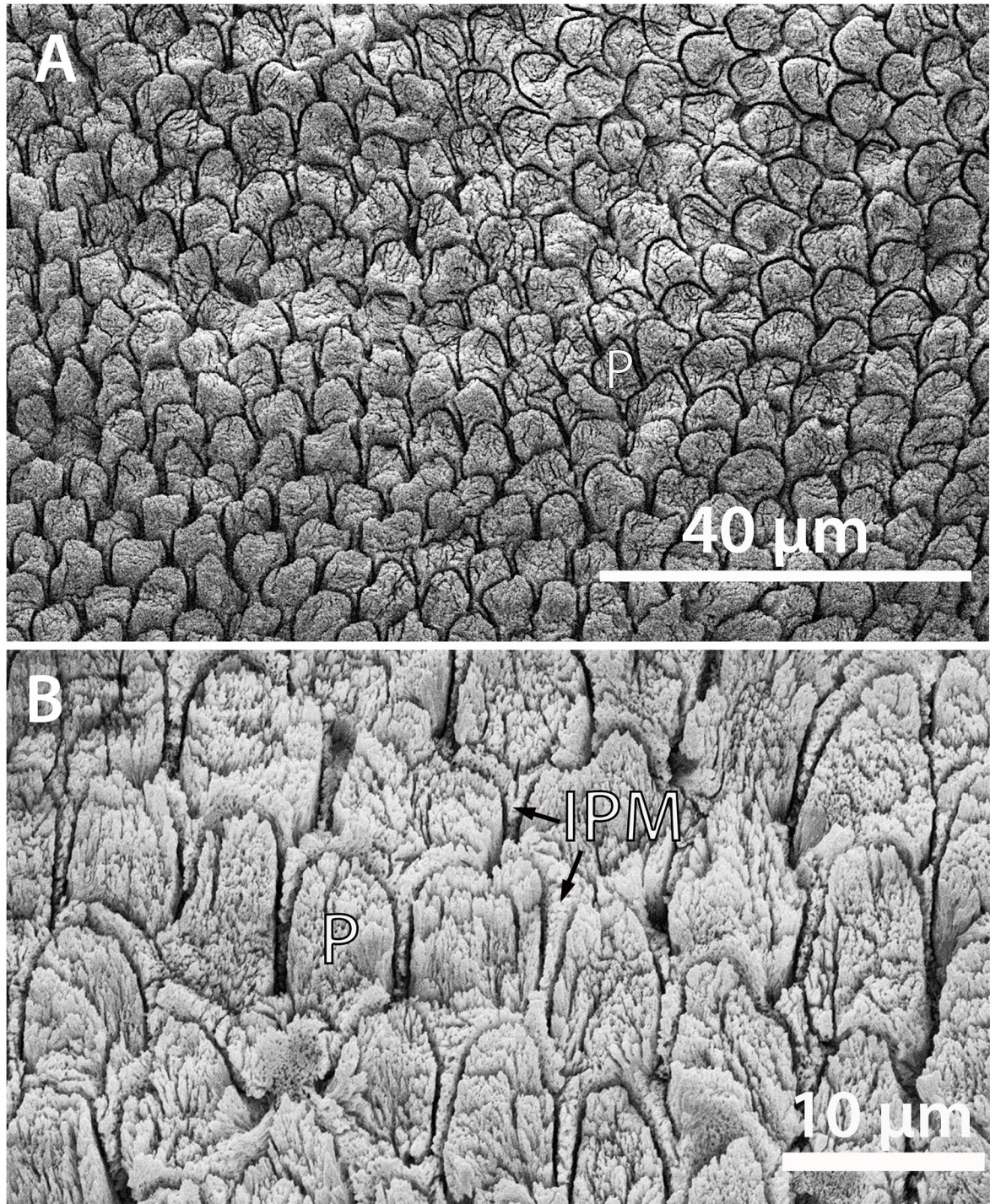

**Fig 6. SEM images of the tangential section of NUFV 2092E.** (A) Cross-section through prisms; the prism sheath is dark and looks like a trench around each prism because it has been etched away. (B) Close-up of prisms and interprismatic matrix which are comprised of fine, needle-like crystallites. P, prism.

together, the prism sheaths do not appear to touch one another. Rather, a thin (~1–2 μm) layer of IPM surrounds the prism sheaths and appears as a 'tail' below each prism that tapers towards the base of the tooth or to the side (Fig 6B).

## Schmelzmuster

The combined interpretation of the transverse, vertical, and tangential sections of NUFV 2092B and 2092E indicates that the enamel microstructure is dominated by elongate, steeply dipping or near vertical bodies of prisms that penetrate the enamel almost from the EDJ towards the OES, leaving thin inner and outermost zones.

Next to the EDJ, there is a thin inner zone of radial enamel in some places. However, in other areas next to the EDJ the prisms do not appear to run parallel to one another, and it is difficult to discern a pattern.

The middle zone, which comprises the greatest thickness of enamel, is made up of a complex enamel wherein elongate bodies of variable thickness extend outward from the EDJ towards the OES (Fig 7A). At higher magnification, the elongate bodies are HSB-like in that they are comprised of steeply-dipping or rising prisms, and are separated from one another by transitional zones of horizontal (or nearly so) prisms (Fig 7B). The nearly vertical orientation of the elongate bodies becomes obvious in the tangential section (discussed below). However, in contrast to HSB that are identified in many large mammals [9], the elongate bodies and intervening transitional zones are not of uniform thickness or spacing from one another in NUFV 2092B.

In the uncoated tangential section of NUFV 2092E (Fig 8), the vertical bodies are clearly visible and complex. When light is illuminated from the bottom (Fig 8A), one sees a pattern of straight to slightly wavy, broad white lines that when magnified (Fig 8B), look like nested chevrons whose points are directed vertically or somewhat diagonally up towards the occlusal surface. These lines of nested chevrons or treelike structures extend upwards for a considerable distance, sometimes bifurcating, and are separated from one another by darker bands whose structure is the same, but point in the opposite direction. In the top one-third of the tooth, the light-colored bands become broader, more rounded, and lose their nested chevron appearance, and the darker bands in between are considerably narrower than the light-colored bands. The vertical structures illuminated in the tangential section (Fig 8) appear to parallel the vertical crenulations and wrinkles on the outside surface of the enamel of NUFV 2092E (Fig 4). Further, just as the nested chevron structures transition into broader, more rounded bands towards the occlusal surface of NUFV 2092E in the tangential section, so do the crenulations on the outside surface of the enamel (compare Figs 4B with 8).

NUFV 2092E (Figs 4B and 8), probably a fragment of an anterior premolar, was ground the deepest in the lower third of the tooth, whereas the upper two-thirds of the specimen was more shallowly ground. Therefore, the lower part of the tooth (where damage is evident by cracks and a small hole in the enamel) is hypothesized to capture the microstructure of the inner zone of enamel (nearest the EDJ), whereas much of the area above (external to) the EDJ (Fig 8A) is likely correlative with the thick middle zone that is comprised of vertical, elongate bodies of prisms. The structures extend to the occlusal surface (top) of NUFV 2092E, which is worn, so we hypothesize that the thin outer zone of radial enamel that occurs elsewhere next to the OES has been removed by occlusal wear. In the region of enamel along the left and right margins of NUFV 2092E next to the OES (Fig 8A), however, the microstructure is indistinct and blurred in appearance; this may represent an outermost zone of radial enamel (discussed below).

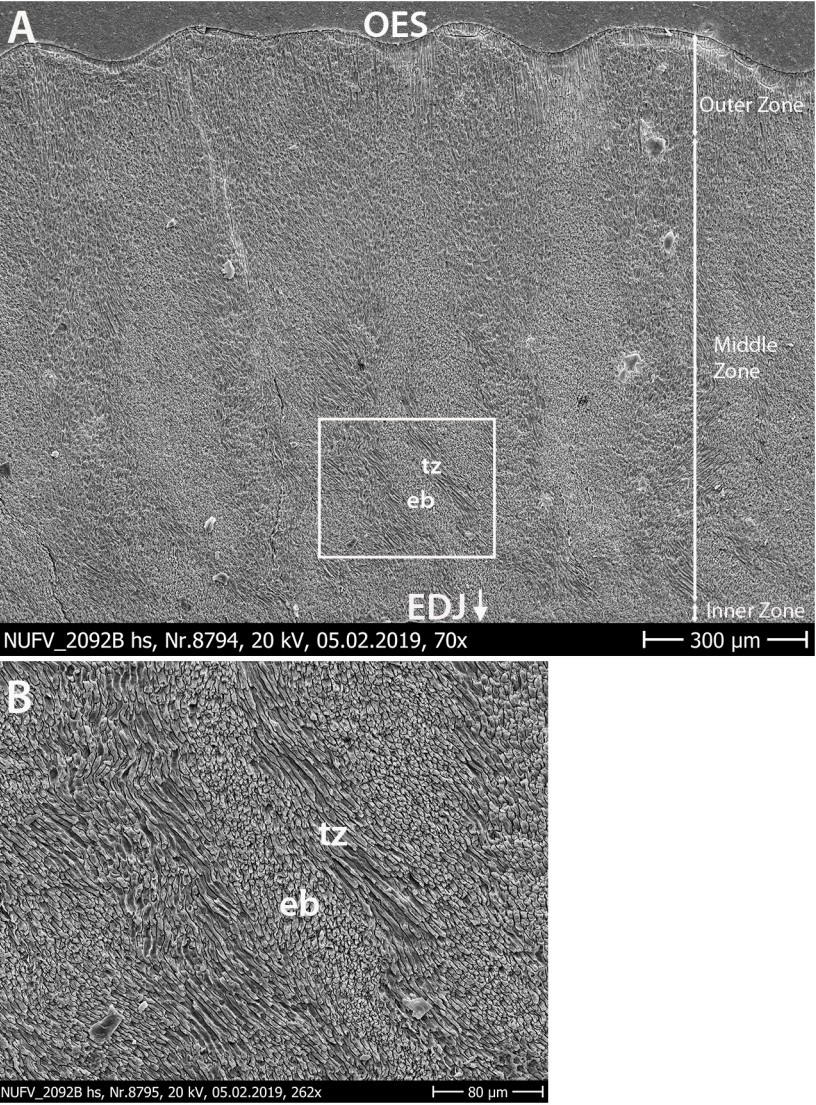

**Fig 7. SEM image of enamel of NUFV 2092B in horizontal section.** (A) Three enamel zones are evident—a very thin Inner Zone of radial enamel (in some places), a thick Middle Zone comprised of elongate bodies of prisms, and a thin Outer Zone. (B) Higher magnification SEM image showing area outlined by white square in (A); elongate bodies (eb) are comprised of steeply-dipping or rising prisms, and are separated from one another by transitional zones (tz) made up of horizontal (or nearly so) prisms. Irregular-shaped grains on SEM images are an artifact of sample preparation process.

Based on the tangential section (Fig 8), we predict that the nested chevrons should appear in cross section as vertical elongate bodies with variable thicknesses and branches, depending upon where the horizontal section transects them. Further, we predict that they should be concentrated in the inner region of the middle zone of enamel. Whereas, in the outer region of the middle zone (towards the OES) in horizontal section, we hypothesize that the elongate bodies should be broader in appearance, and the nested chevron structures should all but disappear in the outermost zone of enamel. In fact, the horizontal section (Fig 7) appears to show just that—the outer region of the middle enamel zone shows broader bodies that transition in places into radial enamel near the OES.

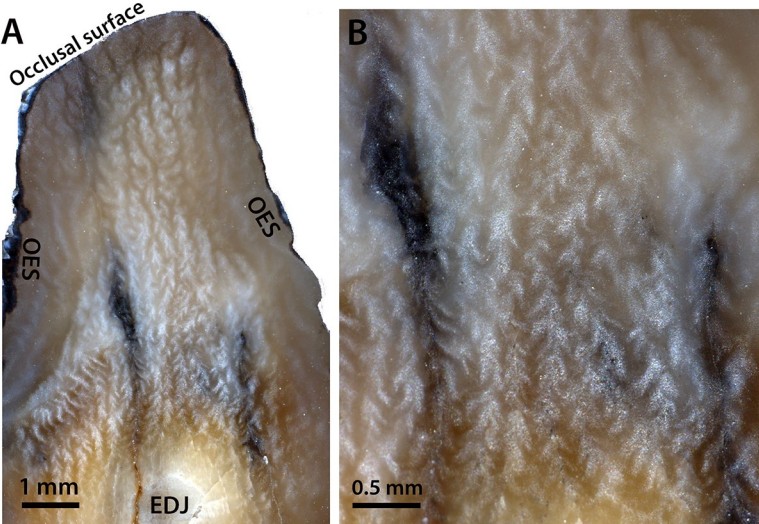

**Fig 8. Uncoated tangential section of NUFV 2092E under a light microscope (A) and higher magnification (B).** In both images, occlusal surface is towards the top and source of light is from the bottom. EDJ, Enamel-Dentine Junction; OES, outer enamel surface.

Evident in vertical sections of NUFV 2092B, an outer zone of radial enamel in which the prisms run parallel to each other lies near the OES (Fig 9A). In some areas, it may be up to 300 μm in thickness, whereas in other areas it is much thinner, and the microstructures of the thick middle zone extend nearly to the OES. In some areas adjacent to the OES, the prism sheath vanishes between the parallel-oriented prisms, so that these merge into a prismless outer enamel zone (PLEX; Fig 9B), a feature observed in a number of mammalian taxa [34–36]. In other areas, the enamel contains large vacuities, probably the result of diagenesis (Fig 9A, bottom left).

In summary, the analysis of NUFV 2092B and 2092E in horizontal, tangential, and vertical sections at both low and high magnification indicates that the enamel is characterized by elongate, vertical bodies extending from near the EDJ towards the OES that are made up of steeply-angled prisms that decussate with adjacent bodies. The inner region of the middle enamel zone contains nested chevron or treelike structures that are evident in horizontal and tangential sections, whereas in the outer region, the vertical bands become broader and the nested chevron structures are lost. An outer zone of radial enamel or PLEX occurs next to the OES.

## Discussion

The enamel microstructure that occurs in NUFV 2092B and 2092E is comparable to that of the pantodont *Coryphodon* described by [19]. Coined "*Coryphodon*-enamel" by these authors, this enamel type is characterized by vertical or oblique structures that in tangential section appear as light-colored bands of nested chevrons or treelike structures separated from one another by similar, though narrower, dark-colored bands. The difference in width between the light and dark stripes when light is reflected from one direction indicates that the prism orientation is not symmetrical [19]. Also like *Coryphodon*-enamel (although not restricted to it), NUFV 2092B and 2092E have an outermost zone of radial enamel, as well as rounded prisms that open to one side and are surrounded by interprismatic matrix that is nearly parallel to the

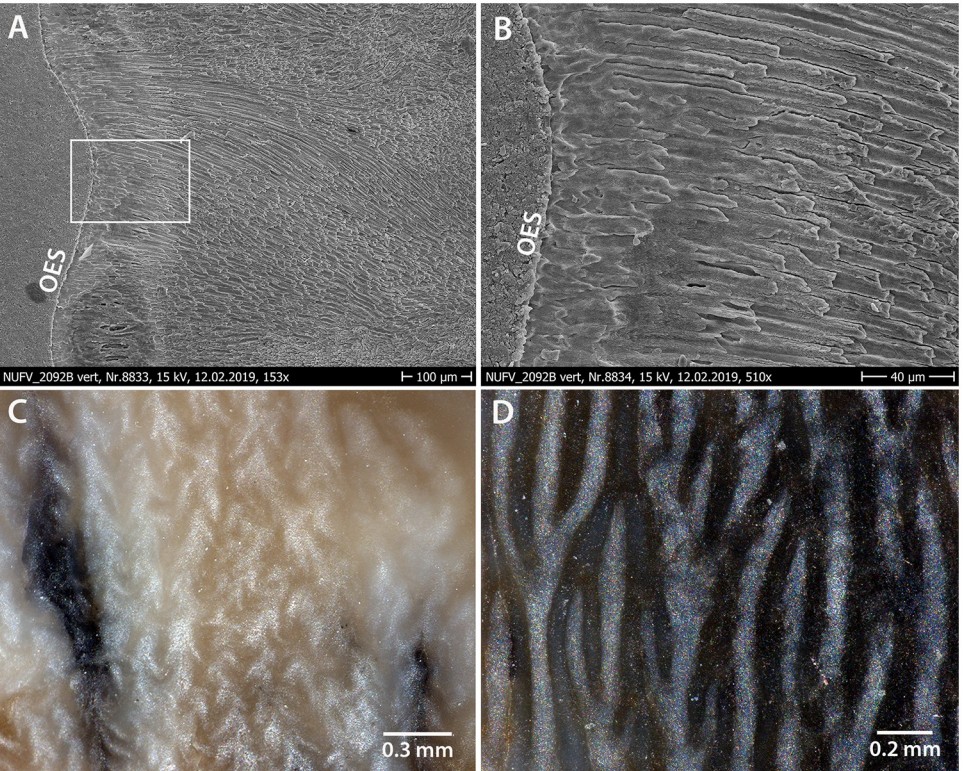

**Fig 9. SEM images of vertical section of outer zone of NUFV 2092B (A and B), and uncoated tangential sections of NUFV 2092B (C) and Yukon Government (YG) 514.12 (D), rhinocerotoid enamel from [26].** In (A) the region outlined by white square is magnified in (B). OES, outer enamel surface. (C) Magnified uncoated tangential section of NUFV 2092E under a light microscope, compared to (D) uncoated tangential section of YG514.12, enamel of a Miocene rhinocerotoid from the Yukon, Canada under a light microscope.

prisms. Given the suite of characters shared with *Coryphodon*-enamel, but predominantly the vertical bodies that manifest as lines of nested chevrons or treelike structures in tangential view, NUFV 2092B and 2092E possess *Coryphodon*-enamel. However, does *Coryphodon*-enamel occur in any of the other large mammals known from the Arctic localities?

In the Canadian Arctic, rhinocerotoids are known from early Miocene sediments of the Haughton Formation on Devon Island (Dawson 1990) and from the Yukon [26]. Rhinocerotoids have vertical HSB in their tooth enamel that look somewhat like the vertical elements in *Coryphodon*-enamel [9, 23, 37]. However, the vertical bands in rhinocerotoid enamel contrast with those in *Coryphodon*-enamel in that they are of consistent thickness and spacing from one another and separated by thin transitional zones of 2–3 prisms wide. In tangential section, rhinocerotoid enamel shows light and dark vertical lines that bifurcate in a regular pattern and lack the nested chevron or treelike structures in *Coryphodon*-enamel (Fig 9C and 9D).

Although not nearly as regular in pattern as the vertical HSB in rhinocerotoids, *Coryphodon*-enamel nevertheless cannot be considered irregular enamel. This enamel type, which occurs in proboscideans (elephants and their extinct relatives) and some rodents, is characterized by prisms that twist irregularly in bundles or as single prisms around each other, and bundles of prisms show a range of angles and attitudes with no consistent pattern [18, 38].

The brontotheres cf. *Eotitanops* and *Palaeosyops* are documented from early—middle Eocene rocks of the Margaret Formation on Ellesmere Island [14, 32], and tooth fragments

from a larger, younger brontothere were recovered from middle Eocene strata of the Buchanan Lake Formation on nearby Axel Heiberg Island [39]. However, brontothere tooth enamel differs from *Coryphodon*-enamel in having U-shaped HSB, an intermediate condition between the horizontal HSB found in most large mammals and the vertical HSB of rhinocerotoids [23, 40]. Tapiroids occur at early Eocene localities in the Margaret Formation [14, 41]. However, tapiroids have horizontal to curved HSB [9, 38].

The nested chevron pattern seen in the tangential section of *Coryphodon*-enamel is reminiscent of the zigzag HSB found in advanced carnivorans, particularly *Crocuta crocuta* (hyaena) that correlate with ossiphagous (or bone-eating) habits [19, 42]. However, the enamel of *C. crocuta* differs from *Coryphodon*-enamel in that the vertical structures show a symmetrical pattern when illuminated from opposing directions in tangential section, and the vertical light and dark bands are of similar thickness [19, 42].

Several Carnivoromorpha are known from the Margaret Formation [43], but their enamel shows significant differences from *Coryphodon*-enamel. *Miacis*, cf. *Vulpavus*, and *Viverravus* are small-bodied members of Carnivoromorpha whose tooth enamel is much thinner (and smoother) than that of NUFV 2092B and 2092E. Further, the tooth enamel of these early Eocene carnivores contains undulating HSB, which are essentially horizontal, slightly wavy HSB [42]. The oxyaenid *Palaeonictis* and mesonychid *Pachyaena* are the largest carnivores in the Margaret Formation, although their fossils are rare, with each taxon represented by a single fossil in the Arctic [43]. The enamel of these taxa has undulating HSB in the lower half of the tooth that transitions into zigzag HSB in the upper one-half to one-third of the tooth [44]. *Coryphodon*-enamel altogether lacks undulating HSB. In addition to its namesake, *Coryphodon*-enamel is known to occur in middle Eocene *Uintatherium* and late Eocene *Entelodon* [19], neither of which is known from the early Eocene nor from the polar region. Therefore, it is unlikely that the *Coryphodon*-enamel reported here from the Strathcona Fiord Fossil Forest belongs to any other mammal besides *Coryphodon*.

## Conclusions

The tooth enamel fragments reported here, along with some poorly preserved bone fragments, thus far are the only documented vertebrate fossils from the Strathcona Fiord Fossil Forest. However, *Coryphodon* fossils occur elsewhere in the Margaret Formation on Ellesmere Island [30], so its presence at the Strathcona Fiord Fossil Forest is not surprising. What is novel in our study is the way in which we identify the fossils NUFV 2092B and 2092E to *Coryphodon* by way of their enamel microstructure. Complete mammalian teeth and jaws are morphologically diagnostic and readily identified to genus and even species. However, in the Arctic, Eocene vertebrate fossils are rare, and many are fragmentary. Here, we provide an example of how enamel microstructure can be used to identify the mammal.

The presence of *Coryphodon* suggests that the strata containing the Strathcona Fiord Fossil Forest are temporally correlative with the early Eocene (Wasatchian) fossil-bearing strata of the Margaret Formation approximately 25 km further north at Bay Fiord, a conclusion reached independently by lithologic correlation. At Bay Fiord, perissodactyls, hyaenodontid creodonts, *Miacis*, and cf. *Vulpavus*, all of which first appear at mid-latitudes in the Wasatchian, as well as the Wasatchian index taxon *Pachyaena* and the archaic ungulate *Anacodon*, which last appears in the Wasatchian, occur in the lower faunal level of the Margaret Formation [1, 2, 3, 14]. Although *Coryphodon* occurs at early middle Eocene (Bridgerian) localities at mid-latitudes, it is restricted to the early Eocene (Wasatchian) faunal assemblage in the Arctic [30]. The strata containing the Strathcona Fiord Fossil Forest were initially mapped by Harrison et al. (2009) as the Late Paleocene Mount Lawson Formation of the Eureka Sound Group. However, our

study indicates that they should be re-mapped as the Margaret Formation on the basis of lithology, paleoenvironmental interpretations, and the presence of *Coryphodon*.

Vertical elements such as those found in the enamel of the carnivore *Crocuta* (hyaena) and entelodonts are hypothesized to be an adaptation to deal with high stresses during mastication [19, 44]. *Crocuta* is carnivorous and ossiphagous, whereas entelodonts have been interpreted as omnivores and pig-like in their diet, ingesting a variety of food items, including plants, meat, and bones [45, 46]. In contrast, *Coryphodon*, based on its transverse shearing lophs and carbon isotope values, is interpreted as an herbivore, and probably semiaquatic [47]. Based on its oxygen and carbon isotope values, *Coryphodon* is inferred to be a year-round resident above the Arctic Circle and therefore experienced months of darkness and the shutdown of photosynthesis during the polar winter [48]. Seasonal isotopic variations, and specifically high winter $\delta^{13}C$ values in the enamel of *Coryphodon* teeth from Ellesmere Island, suggest a varied diet during the dark winter that probably included wood, leaf litter, and evergreen conifers [48]. The presence of vertical elements in the enamel microstructure of *Coryphodon* may have pre-adapted these large mammals to ingesting tough, poorer quality food items during the long dark winters above the Arctic Circle. This could partly explain why *Coryphodon* is the most abundant herbivore in the Eocene Arctic.

## Acknowledgments

We thank colleagues at the Nunavut Government (A. Stubbing, S. LeBlanc) and the Canadian Museum of Nature (K. Shepherd, M. Currie) for loan of the fossils. Enamel microstructure analysis of the tooth fragments was carried out by J. Eberle and W. v. Koenigswald in the Pale-ontology Section of the Institute of Geosciences at the University of Bonn in Germany. We thank T. Martin, Chair of Palaeontology at the Institute of Geosciences at the University of Bonn, for allowing use of his enamel and imaging labs for preparation and imaging of the fos-sils. We also thank G. Oleschinski (Palaeontology, University of Bonn) and H.-J. Ensikat (Institut für Biodiversität der Pflanzen, University of Bonn) who worked with us to produce the SEM images. Careful reviews by K. Rose, K.C. Beard, an anonymous reviewer, and Aca-demic Editor A. Csank improved the manuscript. Publication of this article was funded by the University of Colorado Boulder Libraries Open Access Fund.

## Author Contributions

**Conceptualization:** Jaelyn J. Eberle, Wighart von Koenigswald.

**Formal analysis:** Jaelyn J. Eberle, Wighart von Koenigswald, David A. Eberth.

**Investigation:** Jaelyn J. Eberle, Wighart von Koenigswald, David A. Eberth.

**Methodology:** Jaelyn J. Eberle, Wighart von Koenigswald, David A. Eberth.

**Resources:** Jaelyn J. Eberle.

**Writing – original draft:** Jaelyn J. Eberle, Wighart von Koenigswald, David A. Eberth.

**Writing – review & editing:** Jaelyn J. Eberle, Wighart von Koenigswald, David A. Eberth.

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
