## [Decision Letter · Decision Letter 0]

24 Jun 2020

PONE-D-20-10764

Using tooth enamel microstructure to identify mammalian fossils at an Eocene Arctic forest

PLOS ONE

Dear Dr. Eberle,

Thank you for submitting your manuscript to PLOS ONE. After careful consideration, we feel that it has merit but does not fully meet PLOS ONE’s publication criteria as it currently stands. Therefore, we invite you to submit a revised version of the manuscript that addresses the points raised during the review process.

This is an excellent study as noted by all three reviewers and although the system does not permit a category for 'very minor revisions' that is what these are. If the authors can make the suggested changes, which are primarily along the lines of clarifying a few things. I will be happy to accept the manuscript in very short order.

We look forward to receiving your revised manuscript.

Kind regards,

Adam Csank, Ph. D.

Academic Editor

PLOS ONE

Journal Requirements:

2. Please add the permit details to the Material and Methods section of your manuscript.

Additional Editor Comments (if provided):

Reviewers' comments:

Reviewer's Responses to Questions

**Comments to the Author**

1. Is the manuscript technically sound, and do the data support the conclusions?

Reviewer #1: Yes

Reviewer #2: Yes

Reviewer #3: Yes

2. Has the statistical analysis been performed appropriately and rigorously? 

Reviewer #1: N/A

Reviewer #2: N/A

Reviewer #3: N/A

3. Have the authors made all data underlying the findings in their manuscript fully available?

Reviewer #1: Yes

Reviewer #2: Yes

Reviewer #3: Yes

4. Is the manuscript presented in an intelligible fashion and written in standard English?

Reviewer #1: Yes

Reviewer #2: Yes

Reviewer #3: Yes

5. Review Comments to the Author

Reviewer #1: I preface my remarks by acknowledging that some of my comments may be irrelevant or inaccurate, because I have not been able to check some details in pertinent publications or by examining actual specimens (I do not have access to my library or collections at present, since my university remains closed due to covid-19). Some relevant articles (e.g., book chapters) are not available on line.

Over the last several decades a small group of intrepid vertebrate paleontologists (including two of the three authors) have been exploring the Canadian High Arctic for fossil vertebrates. Though fossils are relatively scarce, their efforts have resulted in discovery of a sizable vertebrate fauna. These fossils are especially important for understanding Eocene paleoclimate and paleoenvironments, climate change, and mammalian dispersal patterns. In this article the authors report the first fossil mammal remains from an Eocene Arctic forest known as Strathcona Fiord Fossil Forest, in the Margaret Fm. of Ellesmere Island. Underscoring the rarity of vertebrate fossils from this fossil forest, the remains consist only of a couple of enamel chips that the authors deem undiagnostic from their external morphology. They provide compelling evidence from study of enamel microstructure (at several magnification levels) that these fragments belong to the well-known Holarctic pantodont Coryphodon (based on previous documentation of Coryphodon enamel). To my knowledge this is the first time anyone has employed enamel microstructure to identify a fossil mammal to genus. It therefore demonstrates the potential of enamel microstructure as a tool for identifying otherwise indeterminate mammal tooth fragments.

The manuscript is clear, well written, and well illustrated. I have no mandatory changes and only a few observations the authors may wish to consider in their final manuscript.

Regarding the described enamel fragments, their size and relatively flat or slightly curved surface alone (indicating derivation from a much larger tooth) already eliminates most early Eocene mammal taxa, including tapiroids, Phenacodus. Size would not necessarily rule out rhinocerotoids (which have somewhat similar enamel), but unless the authors are implying that the fossils are contaminants or that the early Eocene age of the Margaret Fm. is erroneous, rhino is a very unlikely ID. Brontotheres known from the Margaret Fm. have U-shaped HSB, eliminating them. I do not know if the enamel of late early Eocene Lambdotherium (usually considered a basal brontothere) has been investigated, but these fragments may be too big to represent Lambdotherium. Several other taxa that have somewhat similar enamel structure (noted in the text) can be discounted for reasons of geologic age or biogeography.

The authors state (lines 79-81) that “the tooth fragments from the Strathcona Fiord Fossil Forest are so incomplete as to be undiagnostic by using their external morphology,” and this appears to be true from their very fragmentary condition. However, the surface texture visible in Fig. 4 is unusual and characteristic of Coryphodon (as observed by Koenigswald & Rose, 2005); it is quite possibly diagnostic. (If the authors know of any other early Eocene taxon that shows the same surface texture, a figure illustrating this would be useful to demonstrate that these fragments cannot be identified by surface texture alone.) So everything points to Coryphodon—the size, enamel surface features, and previous documentation from the Margaret Fm. all would have suggested this without analysis of enamel microstructure. But the enamel study confirms this ID and shows that enamel microstructure can be a valuable tool for taxonomic determination. The identification of these fragments as Coryphodon is convincing.

To make their conclusions even more compelling, I suggest the following additions—if possible. Could sections of enamel from undoubted Margaret Fm. Coryphodon be examined and compared with the tooth fragments described here? The only other large mammals present, or possibly present, in the Margaret Fm., whose enamel might be confused with that of Coryphodon are oxyaenid creodonts and mesonychids, which have somewhat similar zigzag enamel (Stefen, 1997). The authors state that both have a different pattern, with undulating HSB in the lower part of the tooth and zigzag enamel above. Would it be possible to include an illustration showing how they differ from the Coryphodon tooth fragments? (The large oxyaenid Oxyaena is a common early Eocene mammal that could potentially be present in the Margaret Fm., though not yet recorded—it should be eliminated as well.)

Geologic age of the enamel fragments (early Eocene) is one piece of evidence supporting their identification as Coryphodon, but it is circular reasoning to then cite this as evidence that the Margaret Fm. is early Eocene in age. Is there independent evidence of the age of the Margaret Fm.? “Lithologic correlation” is cited but not further explained. Are any volcanics or datable minerals present, any paleomag or isotope stratigraphy, any age-diagnostic pollen present, or nannoplankton in the underlying marine beds? Or is the early Eocene age based mainly on the mammalian fossils, including the identification of these fragments as Coryphodon? The end of the second paragraph of the conclusions lists presence of Coryphodon as a reason to (re)interpret the Margaret Fm. as early Eocene rather than late Paleocene as originally mapped by Harrison et al. (2009); but as the authors note, Coryphodon has been reported from earliest Bridgerian (Br1a; Gunnell et al., 2009), and it is also known from the latest Paleocene (Clarkforkian, Rose, 1981). If other age-diagnostic mammals are known from the formation, it would be worth listing them here.

At the end of the Conclusions, the authors suggest that its enamel pattern preadapted Coryphodon for eating a tough diet of wood and evergreens during the Arctic winter, and that this might explain its abundance in the Arctic while equoids and artiodactyls are absent there. Maybe so, but if Coryphodon evolved in the mid-latitudes where it is abundant, why did it develop this bizarre enamel pattern in the first place? Or are the authors speculating that it evolved in the Arctic? Given the abundance of equoids and artiodactyls in the early Eocene of both North America and Europe (and the likelihood that intercontinental dispersal occurred during the PETM), it seems that both had to have been present in the Arctic but have not yet been discovered.

Minor issues:

Please define NUFV, YG, and repository (if different).

Line 316—should read (Fig. 4B and 8)

Line 320—‘area above the EDJ’: ‘above’ is ambiguous; use ‘external to’ or add (i.e., toward the OES

Line 404—replace “creodont” with either oxyaenid (preferable) or oxyaenid creodont

Is it possible to locate Dawson’s previous samples relative to the Strathcone Fiord Fossil Forrest in Fig. 3 (or in the text following line 107)? Is there significance to the colors in Fig. 3?

Fig. 6B, label IPM

Why is there a Supplemental File? It appears to be identical with Fig. 5.

Reviewer #2: This is a clearly written and straightforward contribution that illuminates a novel method for identifying fragmentary remains of fossil mammals for paleobiological research. I have only a few minor edits to propose/suggest:

line 245: "astropotheres" should read "astrapotheres"

lines 278 & 337: "parallel" really isn't a verb, so maybe "run parallel to one another" is better here

line 401: I don't know whether you want to conform to the convention proposed by J. Flynn, F. Sole and others, but these stem members of the carnivoran clade are called Carnivoramorpha by them, i.e., not "true" or crown-clade carnivorans

line 404: There is a current debate about the monophyly of "creodonts" (oxyaenids and hyaenodontids). So, again, it might be safer to refer to Palaeonictis as an oxyaenid just to be safe.

I like your hypothesis about Coryphodon being able to withstand Arctic winters while equids and artiodactyls were not. Nevertheless, most biogeographic models for Laurasian mammals have promoted a Thulean land bridge between Greenland and NW Europe as the most likely pathway for dispersal of equids and artiodactyls between North America and Europe, meaning these clades probably occupied Ellesmere or adjacent areas sometime during the early Eocene. Care to comment on this here?

Reviewer #3: Th manuscript is a good example that fossil fragments are also useful to identify taxa. Enamel microstructure is species specific. The use of several techniques and scales of observation is fruitful, even if some samples are "destroyed" by the preparative processes (polishing, etching). The authors have made the good choice in trying to identify the samples. There are so many fragments in museums that are not known from this point of view because they have to be intact!

6. PLOS authors have the option to publish the peer review history of their article (what does this mean?). If published, this will include your full peer review and any attached files.

Reviewer #1: Yes: Kenneth D. Rose

Reviewer #2: Yes: K. Christopher Beard

Reviewer #3: No

---

## [Author Response · Author response to Decision Letter 0]

4 Aug 2020

Dear Editor Csank,

We sincerely thank you and the three reviewers for editorial recommendations, questions, comments, and feedback on our manuscript entitled ‘Using tooth enamel microstructure to identify mammalian fossils at an Eocene Arctic forest’ that is under consideration for publication in PLOS ONE. On behalf of my co-authors Wighart von Koenigswald and David Eberth, I am submitting a revised manuscript that addresses the concerns. Your recommendations, as well as those of the reviewers, are addressed below.

Editor’s recommendations:

1. We have renamed the files, following the guidelines of PLOS ONE. Specifically, the figures are now Fig1, Fig2, etc. Note: Two of the figures (1 and 6) are revised.

2. Information concerning permits was moved from Acknowledgments to Materials & Methods (first paragraph, bottom of p. 7 – top of p. 8).

3. Co-author David Eberth rebuilt Figure 1 (the map) so that it no longer contains a satellite image with copyright issues. The map is simpler, but conveys the same information contained in our original Figure 1 that had the satellite image. I believe we have removed the copyright concerns.

4. There are no Supporting Information files for our manuscript. I apologize, but I may have inadvertently uploaded a figure from the manuscript as a Supporting Information figure. Reviewer Ken Rose caught this and commented on it.

Reviewer 1, Ken Rose:

Dr. Rose had no mandatory changes to the manuscript. However, he made some recommendations.

1. Dr. Rose indicated that the identification of the NUFV tooth enamel frags as Coryphodon was convincing. However, to make the argument even stronger, Ken asked whether sections of enamel from undoubted Margaret Fm. Coryphodon could be examined and compared with the tooth fragments described here (if possible).

Response: Eberle returned the Coryphodon material on loan from the Canadian Museum of Nature (CMN) and Nunavut Government. Due to the COVID-19 pandemic, the CMN is closed (until Nov/Dec, I have been told). Further, the laboratory instrumentation (SEM) needed to analyze the Arctic Coryphodon material is not available to Eberle at CU-Boulder, due to limited access to labs as a result of the COVID-19 pandemic.

2. Dr. Rose also suggested that, if possible, it would be nice to illustrate the enamel microstructure of an oxyaenid creodont and a mesonychid, as both are large mammals, and they have somewhat zigzag enamel according to Stefen (1997).

Response: Co-author von Koenigswald, who was Stefen’s PhD advisor, had hoped to re-analyze the enamel of an oxyaenid creodont and mesonychid at the University of Bonn this summer (in response to Ken’s suggestion). However, Dr. von Koenigswald has limited access to the SEM lab and the enamel collection due to the COVID-19 situation; further, the SEM utilized for our study at the University of Bonn is not currently working and will not be repaired until the COVID-19 pandemic dissipates. Therefore, we are unable to include a figure of the enamel microstructure of an oxyaenid creodont and mesonychid. However, having worked with Stefen, Dr. von Koenigswald is familiar with the enamels of these mammals, and he is responsible for much of the descriptive text on the bottom of p. 18 and top of p. 19. 

3. Dr. Rose suggested that we include other age-diagnostic mammals in the Conclusions section that support a Wasatchian age for the Margaret Formation. We have done so (bottom of p. 19 – top of p. 20). There are no ash ages from the Margaret Formation at Bay or Strathcona fiords, although there is an ash age from the Margaret Formation at Stenkul Fiord (southern end of Ellesmere Island) that yielded a preliminary date of 52.6 + 1.9 Ma (Reinhardt et al. 2010), which fits with the late Wasatchian age suggested by mammals. We did not include this ash age in the manuscript because it’s from a meeting abstract (and considered preliminary), it is unclear how it ties into the fossil vertebrate localities at Stenkul Fiord, and there are questions concerning how the Stenkul Fiord section correlates with the sections at Bay and Strathcona Fiords on central Ellesmere. The fossil mammals are the best means for dating the Margaret Formation.

4. Both reviewers Rose and Beard had concerns about the phrase in our final sentence of the manuscript…. ‘whereas early Eocene artiodactyls and equids, whose teeth are bunodont and not at all adapted to chewing tough plant parts [47], are absent from the Eocene Arctic.’ It is a mystery as to why these two clades haven’t been found in the Eocene Arctic, given their abundance at early Eocene mid-latitude localities, the hypothesized Thulean land bridge that is also supported by the tectonic framework (summarized by Eberle and Greenwood, 2012), and the fact that paleontologists have been searching for fossil mammals on Ellesmere since 1975. To satisfy the reviewers’ concerns, we simply removed the phrase from the manuscript. However, the absence of horses and artios from the Arctic is not for a lack of looking for them! Maybe these mammals inhabited environments that are not preserved today in the Arctic. This is arm waving though. I do not think we have a satisfactory answer (yet). 

5. Dr. Rose had several minor spelling/grammatical issues that we corrected in the manuscript. Also, as per the reviewer’s recommendation, we revised Figure 6 to label the IPM.

6. Dr. Rose asked whether it is possible to locate Dawson’s previous samples relative to the Strathcona Fiord Fossil Forest on Figure 3. Good question! Unfortunately, we cannot. Dawson’s original localities (see Dawson et al., 1976) are approximately 10 – 12 km from the Strathcona Fiord Fossil Forest. In July 2018, Eberle and Kirk Johnson (Smithsonian Director) were filmed for a documentary at the Strathcona Fiord Fossil Forest, and attempted to reach Dawson’s original localities in the Strathcona Fiord area on foot. However, poor weather precluded us from getting to them (or the general vicinity of where they are mapped by Dawson et al., 1976).

7. Dr. Rose is correct. There is no Supplementary File. I apologize – I suspect that I inadvertently submitted one of the figures as a supplementary file.

Reviewer 2, Chris Beard:

1. Dr. Beard’s recommendations are minor – we have corrected all of the spelling/grammatical suggestions made by him. We replaced Carnivora with Carnivoramorpha, and we have replaced ‘creodont’ with oxyaenid (in reference to Palaeonictis), given the hypothesized paraphyly of “Creodonta.”

2. As noted above, we have removed the phrase at the end of the manuscript concerning the absence (thus far) of equids and artiodactyls in the Eocene Arctic.

Reviewer 3, Anonymous:

 This reviewer had no recommended revisions.

This summarizes our responses to the reviewers’ recommendations. We sincerely appreciate your consideration of our revised manuscript for publication in PLOS ONE! Please let me know if you have additional questions. Note: I will be in the field (and away from email) from August 5 – 9, returning August 10.

Sincerely,

Jaelyn Eberle

Curator of Fossil Vertebrates, University of Colorado Museum of Natural History

Professor, Geological Sciences, CU-Boulder

---

## [Editor Report · Decision Letter 1]

31 Aug 2020

Using tooth enamel microstructure to identify mammalian fossils at an Eocene Arctic forest

PONE-D-20-10764R1

Dear Dr. Eberle,

We’re pleased to inform you that your manuscript has been judged scientifically suitable for publication and will be formally accepted for publication once it meets all outstanding technical requirements.

Kind regards,

Adam Csank, Ph. D.

Academic Editor

PLOS ONE
---

## [Editor Report · Acceptance letter]

9 Sep 2020

PONE-D-20-10764R1

Using tooth enamel microstructure to identify mammalian fossils at an Eocene Arctic forest

Dear Dr. Eberle:

I'm pleased to inform you that your manuscript has been deemed suitable for publication in PLOS ONE. Congratulations! Your manuscript is now with our production department.

Kind regards,

on behalf of

Dr. Adam Csank 

Academic Editor

PLOS ONE